



# Fuel types and fire severity effects on atmospheric pollutant emissions in an extreme wind-driven wildfire

Albert Alvarez[1*], Judit Lecina-Diaz [2], Miquel De Cáceres[1], Jordi Vayreda[1], Javier Retana[1,3]

[1] CREAF, E08193 Bellaterra (Cerdanyola del Vallès), Catalonia, Spain

[2] Technical University of Munich, TUM School of Life Sciences, Ecosystem Dynamics and Forest Management Group, Hans-Carl-von-Carlowitz-Platz 2, 85354, Freising, Germany.

[3] Universitat Autònoma de Barcelona, E08193 Bellaterra (Cerdanyola del Vallès), Catalonia, Spain

*Correspondence to*: Albert Alvarez Nebot (a.alvarez@creaf.uab.cat)

**Abstract**

In the Mediterranean area, wind-driven wildfires with crown fires are rising, causing an increment in atmospheric pollutant emissions. Quantifying gas emissions in these wildfires requires a better understanding of the components that contribute to the total emission estimate. Here, we aimed to analyze the differences in pre-fire available biomass distribution among layers of fuel types in *Pinus halepensis* and *Quercus suber* (hereafter, pine and oak) forests burned in one of the largest wildfires

("La Jonquera", 10,264 ha) of the past decades. This was done in order to try to unravel the differences in fire severity linked to the percentage of available biomass consumed in each layer and pollutant emissions ($CO_2$, $CO$, $CH_4$, $PM_{2.5}$). We used field data (>100 post-fire plots) in which measures from crown, shrub and litter layers, fire severity and consumption assessments were combined with data from National Forest Inventories to quantify final atmospheric pollutant emissions.

Total pre-fire available biomass among pine and oak forests showed different vertical distribution. Pine forests had a higher

percentage of crown fine and shrub biomass for all fuel types while oak had more litter biomass. The fuel types with large trees and low tree density, together with fuel types with has lower tree density and vertical continuity had the highest non-charred fire severity in pine and oak. The presence of *Erica arborea* caused higher fire severity in oak stands. Fuel types of pine were more resistant to the effects of surface fires because they had taller trees than oak. Percent biomass consumption was higher in pine and oak stands in low fire severities because the taller trees could withstand surface fire at high intensities

without increasing fire severity. The wildfire analyzed was a large fire with massive crown and high-intensity surface fires, but only a small amount of the finest crown biomass and coarse surface fuels were consumed. Fire severity was the main factor determining different amount of emissions without significant influence of fuel types, and only emissions of $CO_2$ and $CH_4$ were higher in pine than in oak in low fire severities. Although remote sensing technologies are extremely useful for biomass and wildfire severity assessments, field data is essential to quantify biomass consumption, atmospheric pollutant emissions

from different fuel types and fuel layers.



## 1 Introduction

Biomass burning by wildfires is a global phenomenon emitting significant quantities of pollutants such as atmospheric gases, aerosols and particulates into the atmosphere with an important impact on global warming and climate (Bowman et al., 2009; Keywood et al., 2013; Knorr et al., 2016). Global average fire emissions were estimated to be 2.2 Pg C yr$^{-1}$ in the period 1997–2016 (van der Werf et al., 2017). At the same time, climate change affects wildfires, directly by increasing drought conditions that affect fire ignition, propagation, frequency and distribution of extreme wildfire events with high-intensity, and indirectly

through its effects on vegetation and fuels (Ruffault et al., 2018; Fernandes et al., 2022; San-Miguel-Ayanz et al., 2018). Wildfire events release carbon mainly in the form of carbon dioxide (C0$_2$) which together with carbon monoxide (CO), and methane (CH$_4$), constitute nearly 95% of wildfire carbon emissions. From these, mostly CO$_2$ and CH$_4$ have the greatest greenhouse influence, while CO is an active trace gas contributing to the secondary formation of ozone (O$_3$) (Pallozzi et al., 2018). Wildfires also release particulate matter, which can cause health issues with different toxicity levels depending on the

location, particle size and composition (Naeher et al., 2007; Kocbach Bølling et al., 2009).

In recent years, large wildfires have repeatedly affected Europe, particularly in southern Mediterranean countries, which had the majority of the annual burned area from 1980 to 2017 (de Rigo et al., 2017; San-Miguel-Ayanz et al., 2018; Fernandes et al., 2022) with extreme years like 2022 in Portugal, France and Spain (Rodrigues et al., 2023). At the country level, wildfire

emissions were only assessed in Portugal (Rosa et al., 2011; Carvalho et al., 2007; Miranda et al., 2009; Fernandes et al., 2022), Italy (Bacciu et al., 2015) and Greece (Lazaridis et al., 2008), usually analyzing specific periods and adapting the Seiler and Crutzen (1980) method. Moreover, previous studies are usually restricted to prescribed fires that do not experience the range of severities of an extreme wildfire (Fernandes et al., 2022; Balde et al., 2023). Consequently, uncertainties associated with the variables used can influence the amount of material emitted by fires, which may be especially important in

heterogeneous Mediterranean forests with different forest structures. The lack of data from wildfires forces the use of general assumptions without contrasted real data that can cause huge inaccuracies (Balde et al., 2023; Fernandes et al., 2022). For instance, emission factors used in these estimates may not be accurate at species level or for individual components (needles/leaves, branches, shrubs, litter) (Poupkou et al., 2014; Pallozzi et al., 2018; Larkin et al., 2014). In the Mediterranean area, where wind-driven wildfires with crown fires are common (Lahaye et al., 2018; Duane and Brotons, 2018), available

biomass or available fuel (defined as the combustible material that will be consumed in a wildfire under specific weather conditions (EPA, 1996) consumed is not correctly estimated, because higher wind speed is associated with lower crown tree consumption but to a complete consumption of shrubs and litter biomass (Jiménez et al., 2013a; Stocks, 1987; Surawski et al., 2016). These sources of uncertainty can be minimized by using field data from wildfires. However, data on fire occurrences and forest stand characteristics are still scarce in southern Europe, and are essential to understand the impact of different forest

structures and forest types on fire emissions estimates, especially under different burning conditions (Campbell et al., 2007; French et al., 2011; Kasischke and Hoy, 2012; Nunes et al., 2019; Balde et al., 2023).

Fire severity is an important determinant of changes between pre-fire and postfire conditions, and is usually defined as aboveground and belowground consumption of organic matter (Garcia-Llamas et al., 2019; Balde et al., 2023). A wildfire can produce different levels of damage depending on the layers burnt and their flammability (Xanthopoulos et al., 2012; Chiriacó

et al., 2013). Field data is thus key to quantify combustion factors for different pools, burn severities and forest species and structures. Forest structures are usually grouped in fuel types, defined as identifiable associations of fuel elements with distinctive species, form, size arrangement, and continuity that exhibit characteristic fire behavior under defined burning conditions (Alvarez et al., 2012a; García-Cimarras et al., 2021; Abdollahi and Yebra, 2023). While most studies recognize the need to know the combustion factors according to fire severity, field data to parameterize these functions is lacking (Balde et

al., 2023; Jiménez et al., 2013b). Understanding how fire severity is related to fuel consumption among different fuel layers is



needed to improve fire emission estimates to quantify the effect of heterogeneous landscapes on wildfire emissions (Campbell et al., 2007; De Santis et al., 2010; Kasischke et al., 2005; Surawski et al., 2016). Using an unprecedented combination of field data and forest inventory data from a large wildfire a Mediterranean area, the main objective of this study is to analyze the distribution of the available biomass to burn, fire severity and fuel consumption, to ultimately quantify the impact of different

forest structures (i.e., fuel types) and forest types on total pollutant emissions. The specific objectives of the study are: 1) To analyze the distribution of available biomass before the fire for the different fuel types and forest types; 2) To unravel the differences in fire severity among fuel types and forest types, and their consequences on the percentage of biomass consumed in each layer; and 3) To analyze pollutant emissions ($CO_2$, $CO$, $CH_4$ and $PM_{2.5}$) across fire severities and fuel types.

## 2 Materials and methods

### 2.1 Study area and pre-fire vegetation

The study area corresponds to the wildfire that occurred in the Jonquera (North east Spain, Girona province, 42º24'59" N - 2º52'29" E) in summer 2012, which burned 10,264 ha. The climate of the region is mainly humid sub-Mediterranean, with an accumulated annual rainfall between 650 mm in the lower altitudes and 1000 mm at higher altitudes, and its lowest

precipitation in July. Mean annual temperature oscillates around 15-16ºC, with July being the warmest month (mean temperatures in July: 23-24 ºC) and January the coldest (8-9 ºC). The study area was located at 35-780 m a.s.l, with slopes between 0-72%. Through the 20th century, the area was one of the most important regions of the cork (*Quercus suber*) industry. Nowadays, this activity is less common, resulting in an increase in forests and artificial surfaces, and a reduction in shrublands (Salis et al., 2019; Badia et al., 2019). Consequently, fuel load and continuity increased, particularly in areas where population

density has decreased.

Sixty per cent of the area burned in 2012 also burned in a previous large wildfire in 1986 (15,000 ha) and a smaller area burned again in 2006 (420 ha). The burned area in 2012 was characterized by continuous forests of cork oak (*Quercus suber*) and Aleppo pine (*Pinus halepensis*) (hereafter, pine and oak plots). Oak forests were located in the northern part of the wildfire area (Fig. 1), on acid soils with forest structures determined by time since the last fire (27 years or more) with high vertical

continuity and horizontal continuity depending on the structure (Schaffhauser et al., 2011). The understory of oak forests was dominated by two groups of species: i) high density of *Erica arborea;* and ii) other dominant species such as *Cistus monspeliensis*, *Cistus salvifolius, Ulex parviflorus, Genista Scorpius, Quercus ilex or Arbutus unedo*. Pine forests were located in the southern part of the burned area (Fig. 1), on sedimentary rocks with diverse forest structures from medium (1,300-3,000 trees/ha) to low density (<1300 trees/ha). The understory of pine forests was dominated by *Quercus coccifera*, *Pistacea*

*lentiscus and Rosmarinus officinalis*, with lower proportion of *Phyllirea latifolia*, *Quercus ilex*, *Rhamnus alaternus* and *Viburnum tinus.* The main herbaceous species in the two forest types was *Brachypodiun retusum.*

### 2.2 Fire description and weather conditions during the fire

The Jonquera wildfire started at 12.54 pm on 22 July of 2012 and it was controlled at 7.48 am on 27 July. The wildfire started

as a wind-driven wildfire characterized by continuous high strong north-west oriented winds (called ''tramuntana''). Wind speeds were between 10-50 km/h and maximum wind gust was 70 km/h. Fire behavior was extreme with massive crown fires, and a high distance of spots (200-400 m) with a maximum of 1km and high fire spread rates (1.8 km/h). From the morning of the second day until the fire was controlled, the wildfire burnt as a topographic wildfire, burning a lower area without extreme fire behavior. When the fire started, the temperature was 24 ºC (el Pertús) and relative humidity was 24% as a consequence of



the drier winds from the north (DARP, 2012). Moisture content was from 19% to 36% during the spread main period and it was not recovered until the afternoon of the second day (>60%) (data from the meteorological stations of Espolla and Cabanes, at 13 and 20 km from the start of the fire). There was a moderate short-term drought according to the Standardized Precipitation-Evapotranspiration Index -SPEI- (Vicente-Serrano et al., 2010), but there was only a slight medium- and long-term drought. Values of moisture content of shrubs fuel were normal for the station with values for *Rosmarinus officinalis* of

69% and for *Cistus Monspeliensis* of 66%. Regarding *P. halepensis*, the moisture content was 105%, 2% lower than the average value for the same period (station of Port de la Selva, located at 30 km from the burned area for the 18th July 2012; Gabriel et al., 2021).

### 2.3 Field plot data and fire severity estimation

One and a half months after the wildfire, we took videos and photographs from two helicopter flights to capture a complete distribution of fire severity. Using that information, we carried out a first assessment of the area burnt in each forest type using the Land Cover Map of Catalonia 2009 (CREAF, 2009). Field sampling started two months after the wildfire. We obtained data from 111 circular plots of 20 m of diameter, 61 oak plots and 50 pine plots. Following Alvarez et al. (2013) we differentiated three types of fire severity plots (understood as the immediate effect of fire from biotic components of the forest)

according to the percentage of trees with different severity: green plots, which had at least 50% green trees, scorch plots, which had at least 50% scorch trees and not more than 25% green trees, and charred plots which had about 100% charred trees. The number of plots was proportional to each forest type and fire severity in the study area. In each of the two main forest types, we established at least ten plots per fire severity category (Fig. 1).



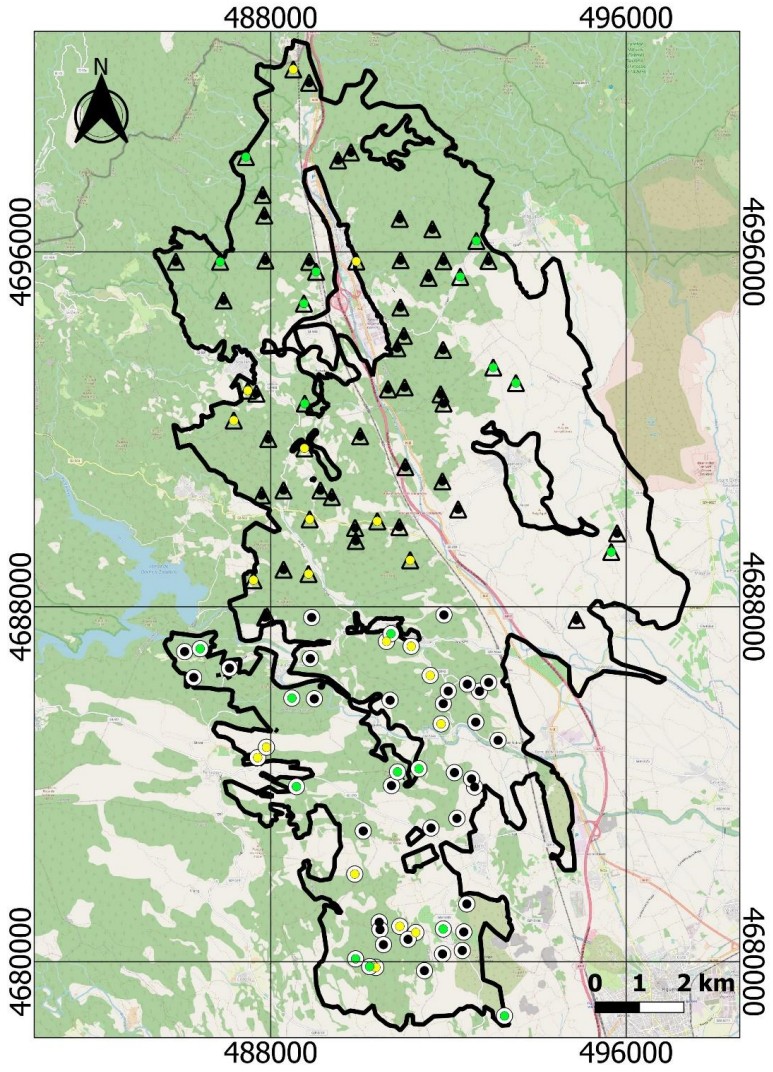


**Figure 1.** Field plots distributed in the fire area (black perimeter). The triangles are the *Quercus suber* plots, the circles are the *Pinus halepensis* plots. The inner dot colors indicate fire severity: low (green), moderate (yellow) and high (black). The base map was sourced from © OpenStreetMap contributors 2024. Distributed under the Open Data Commons Open Database License (ODbL) v1.0. The fire perimeter was provided by The Generalitat de Catalunya firefighters corps.


At the plot level, we measured slope, aspect, elevation and the homogeneity of the forest structure. Following Alvarez et al. (2012a) we considered all trees higher than 3 m independently of species. We classified them in three classes according to their height: small (3–5 m), medium-sized (5–8 m), and large (8 m or taller) to determine forest structure based on their vertical and horizontal continuity. We classified each plot into one of the 20 forest structures in Alvarez et al. (2012a), based on the






number of layers, % of the different types of trees (small, medium, large) and tree density. Subsequently, we grouped these forest structures into 4 fuel types according to their common forest structure characteristics and potential fire type (i.e., active, passive or surface fire) (Table S1 in the Supplement). Fuel type 1 (FT1) was characterized by open forest structures with variable proportions of large trees and very low tree density (<500 trees ha⁻¹), which means very low tree horizontal continuity.

Fuel type 2 (FT2), characterized by large trees and low tree density, was a rare fuel type in oak plots, but it was common in pine plots. Fuel type 3 (FT3) had a similar forest structure than FT2 but with higher tree density and a lower percentage of large trees with a second tree layer. Finally, fuel type 4 (FT4) corresponded to forest structures with a lower than 60% proportion of large trees, high vertical continuity and high tree density (> 1300 trees ha⁻¹) with high horizontal continuity (Fig. 2). This classification of forest structures and fuel types has been successfully applied in several studies related to potential

fire behavior of forest structures (Alvarez et al., 2012a), comparing evolution of the risk of fire type using inventories and to assess the risk of losing ecosystem services (Alvarez et al., 2012b; Lecina-Diaz et al., 2021).

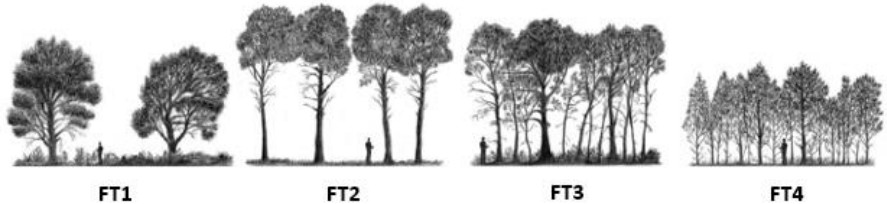

**Figure 2.** Schematic view of the four fuel types (FT1 to FT4) used in this study (based on Figure 2 of Alvarez et al. (2012b)).

For each tree, fire severity was assessed using the proportion of residual crown left alive as an indicator of tree fire severity. Thus, we categorized the tree into three types; firstly, green trees, which could be partially scorch but had at least 20% green

crown; secondly scorch trees, which were mostly affected by radiant and convective heat and had less than 20% green crown, although normally they were fully scorch with abundant fine fuels (needles and small branches with <6 mm) on the tree or on the ground but not consumed; and thirdly charred trees, which were skeletons mainly consumed without fine materials on the tree or on the ground (Alvarez et al., 2013). For each tree we also measured species, diameter at breast height (DBH), total height and crown base height (measured at the lowest part of the crown with vertical continuity of branches and higher and

lower char height measured on tree stem). Moreover, we assessed the percentage of fine fuel consumed of leaves and branches (lower than 0.6 cm) per tree. Regarding shrubs, at the plot level, we identified all shrub species visually and, for each shrub species, we estimated its fraction cover. We also measured visually the percentage of each shrub species consumed, differentiating between particle size classes (i.e., 1-, 10-, 100-, 1000-h time-lag).

**2.4 Computation of pyrogenic emissions**

To assess pyrogenic emissions, we applied the Seiler and Crutzen (1980) method (see a detailed description in Sect. S1) following Eq. (1):

$$EM = A \times B \times C \times D, \tag{1}$$

where EM are the total emissions (Mg/ha), A is the area burned (ha), B is the available biomass before the fire (Mg/ha), C is

the combustion factor (%) and D is the emission factor (g/kg).





### 2.4.1 Area burned and pre-fire available biomass

The area burned was obtained from the fire perimeter measured by the firefighter corps in the field immediately after the wildfire with GPS. At the plot level there were different plot areas depending on the heterogeneity of the area, ranging from 316 to 716 m$^2$. To obtain the fine biomass before the fire we distinguished three components: trees, shrubs and litter.

- For trees, to calculate the available fine biomass before the fire of each tree (leaves and fine branches lower than 0.6 cm) we used the allometric equations considering DBH and height as independent variables to assess biomass (Cáceres et al., 2015) given in the 'medfate' R package, which used those calibrated with data from the Ecological Forestry Inventory of Catalunya (IEFC) (Gracia, 2001).

- For shrublands, we estimated its biomass from field data. First, we chose the closest NFI3 plot to each of our field
plots, in which 70% of the dominant area was either *Q. suber* or *P. halepensis* with the same shrub dominant species found in the field. 83 NFI3 oak plots and 51 pine plots were located within the fire perimeter and county region (Alt Empordà). Then, in these NFI3 plots we calculated the total and fine shrub biomass using the 'medfate' R package (De Càceres et al., 2019). Afterwards, we made a multiple linear regression model (MLR) to obtain allometric equations between total shrub biomass assessed with 'medfate' and shrub cover and dominant shrub species (variables
obtained from NFI3) for oak and pine forests separately (Table S2). From this total shrub biomass, we computed fine shrub biomass with simple linear regression models in each forest type (Table S3). Finally, we applied these allometric equations to the field plots using the shrub cover and dominant species obtained directly in the field to compute total and fine shrub biomass for each field plot.

- To assess the fuel load of litter in each plot, we made allometric equations using previous data from the Ecological
Forestry Inventory of Catalunya (IEFC) (Gracia, 2001). This inventory measured litter biomass, which was not available in NFI3 data (Vayreda et al., 2016). Firstly, we selected the closest IEFC plots to the wildfire with the same dominant tree forest and shrub species found in the field. In this case, we used 91 oak plots and 190 pine plots, located either within the fire perimeter or near it (in the same province, or in close-by provinces for pine data). Afterwards, we applied a stepwise backward method to obtain the best MLR model to assess litter fuel load in the IEFC plots. For
the oak forests, we obtained an MLR in which slope, tree density, shrub cover and time from the last wildfire were the variables explaining litter biomass, while in pine forests the percentage of dominant tree species, total basal area, shrub cover and time from the last fire were the variables that explained litter biomass (Table S4). Finally, we applied those equations to our field data and obtained litter biomass in our field plots.

### 2.4.2 Combustion and emission factors

We considered $CO_2$ and $CH_4$ (two gases that influence the greenhouse effect), CO (an active trace gas that contributes to the secondary formation of ozone $O_3$), and $PM_{2.5}$ (which has been described as an indicator of the hazards to health). The combustion factors (i.e., the proportion of available biomass combusted; Wiedinmyer et al., 2006) for each forest structure and fire severity were obtained from field work. For trees, we made visual estimates in the field on the proportion of leaves and
fine branches (less than 6 mm) remaining on each tree after the fire, then we obtained the % consumed in each tree. We calculated the combustion factor at the plot level for leaves and branches averaging the values of all trees in the plot. For shrubs, we estimated two consumption categories at plot level corresponding to the percentages of fine fuel consumed (i.e., leaves and fine branches <0.6 cm) and the percentage of branches (i.e., > 0.6 cm) consumed. Finally, we also visually assessed the % of litter consumption in the entire plot.



The emission factors, which provide the mass of a compound emitted per mass of dry fuel consumed (Urbanski, 2014), were taken from a combination of emission factors available in the literature for each layer at crown, shrub and ground level. For trees, we used the emissions factors recently applied by Fernandes et al. (2022) from Miranda (2004, 2005), differentiating between pines (other resinous) and oaks. For the shrub layer, in the two forest types we used the same emission factor from Miranda (2005). For litter, we used the values from Pallozzi et al. (2018) available for *P. halepensis*, and we used the *Quercus*

*pubescens* values for *Q. suber* (Table S5). We applied Eq. 1 to obtain the total emissions of $CO_2$, $CO$, $CH_4$ and $PM_{2.5}$ per plot.

**2.5 Statistical analysis**

To determine the differences in available fine biomass among fuel types of oak and pine before the fire, we carried out two analyses. First, we analyzed available biomass between species and fuel types with a two-factor ANOVA with fuel type and

species as factors. Then, to study the vertical distribution of biomass, we computed the percentage of available biomass per layer (i.e., crown, shrub, litter) in each fuel type and species. With these values, we carried out a three-factor ANOVA to analyze the differences in available biomass among layers and fuel types for each species. We applied a log transformation to available biomass to reach normality. After that, we determined the percentage of plots with different fire severity for each fuel type and forest type. Regarding combustion factors, we used field work data and computed two percentages of

consumption for crown trees (leaves and fine branches), two for shrubs (fine fuel and coarse fuel) and one more for litter for each fuel type, fire severity and species. To analyze the differences in emissions of the different compounds ($CO_2$, $CO$, $CH_4$ and $PM_{2.5}$) between forest types across fire severities and fuel types, not all possible combinations of these factors were found in our study area. Therefore, we first analyzed the emissions in green, scorch and charred plots of the two forest types with two-way ANOVAs independently of the fuel types. Then, we analyzed, for each forest type separately, the effect of fire severity

(green, scorch and charred) and fuel type (FT2 and FT3 for pines, and FT3 and FT4 for oaks) on atmospheric pollutant emissions. When the ANOVAs had significant differences, we used the Tukey test to see the levels that were different.

**3 Results**

**3.1 Differences in pre-fire available biomass**

We found significant differences in available fine biomass available to burn before the fire among fuel types (ANOVA test, F= 8.6 p<0.001) and species (F=10.3, p=0.002). The interaction of fuel type and species was also significant (F=3.8, p=0.012). Total fine biomass was higher in oaks than in pines in FT1, with the lowest tree density, and, to a lesser extent FT4, with high vertical and horizontal tree continuity. Biomass was higher in pine than in oak plots in FT2, with dominance of large trees and low density while it was similar among species in FT3 with large trees but a second and higher density than FT2 (Fig. 3).




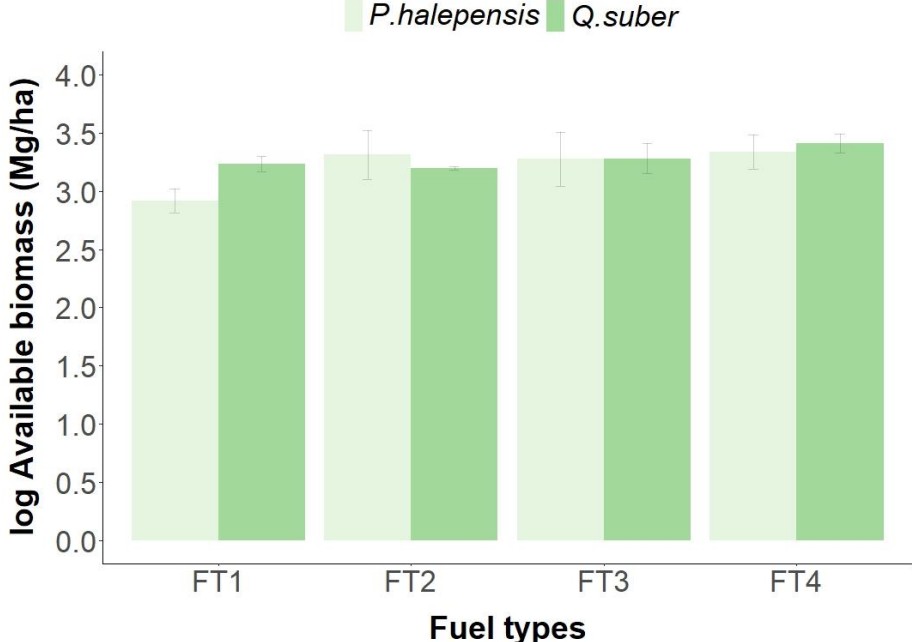

**Figure 3.** Mean (± standard deviation) values of available biomass to burn (Mg/ha) for the different fuel types (FT) for the pine and oak plots. Descriptions of the four fuel types are given in the Methods section.

Available biomass was differently distributed in the different layers in oak and pine plots (Fig. 4). The percentage of the available biomass in the crown was higher in pine than in oak plots (Fig. 4). In oak plots, there was more biomass in the litter than in the shrub layer while in the case of pine plots the distribution was the opposite (more biomass in the shrub layer). Moreover, pine plots had a higher percentage of available biomass in the crown than oak plots. Finally, in pine plots we found differences between the FT1 with the lowest tree density and the other fuel types, which had a higher percentage of shrub
biomass and much less in the crowns in the former than in the other three (Fig. 4).





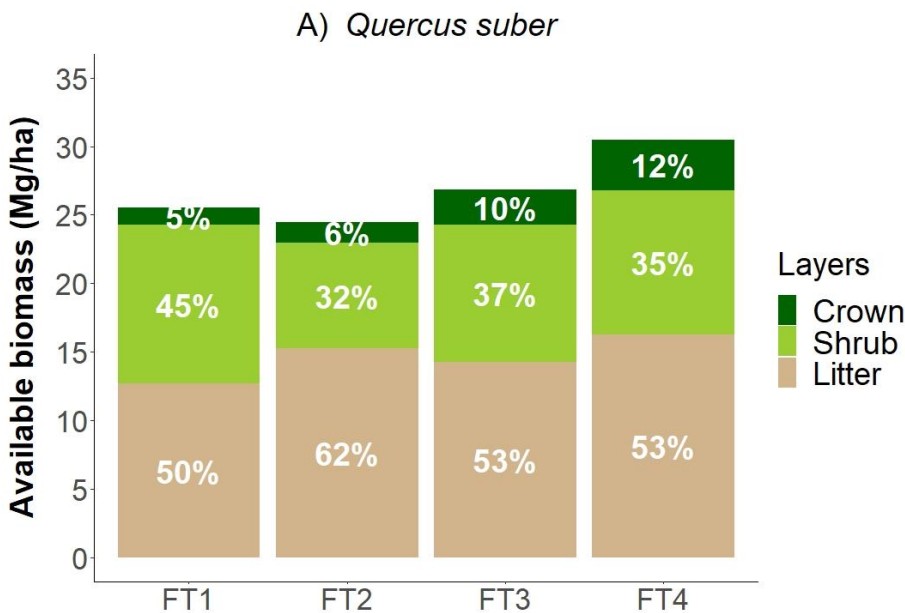

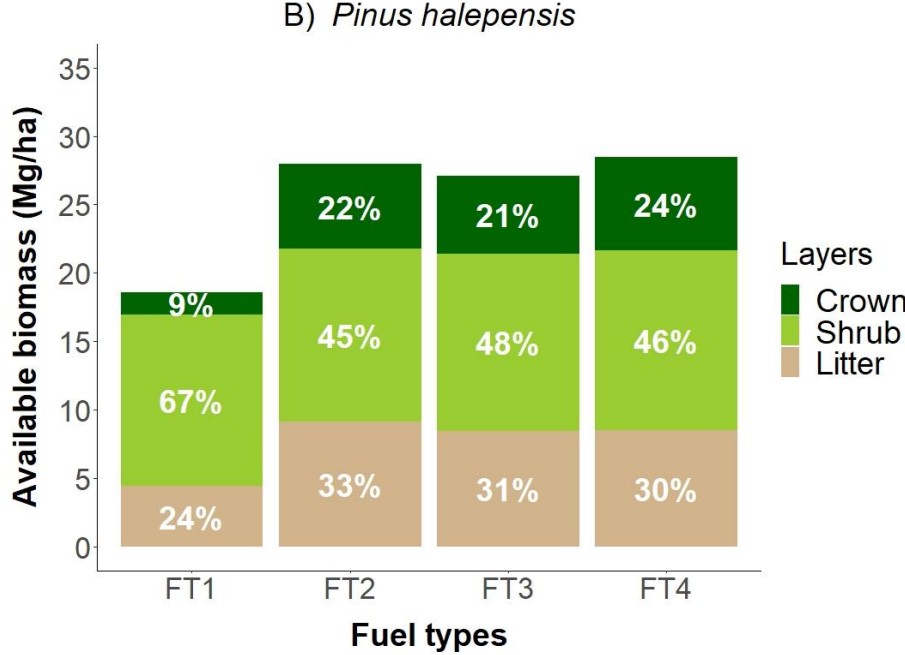

**Figure 4.** Distribution of available biomass (Mg/ha) before the fire in the crown, shrub and litter layers and the four fuel types for A) oak plots; B) pine plots. Numbers in the bars indicate the percentages of available biomass in each layer for each fuel type.



We analyzed the distribution of available biomass among fuel types considering the 3 different layers (crown, shrub and litter) in the two species. The interaction of the three factors was not significant (three-way ANOVA, F=1.4 p=0.18). We found significant differences of available biomass between species (F=4.5 p=0.03), among layers (F=87.2 p<0.001) and among fuel types (F=45.2 p<0.001). The paired interactions between factors were also significant (ANOVA test, p<0.001 in the three cases; see Fig. S1 in the Supplement). Concerning the interaction between species and layer, litter biomass was higher in oak plots, whereas pine plots had more crown available biomass and, to a lesser extent, shrub biomass. The interaction of species and fuel type showed that they had similar available biomass and only pines had a slightly lower biomass in FT1 than in the other three fuel types. Finally, the significant interaction of fuel type and layers indicated that FT2 and FT4 had the highest values of crown biomass and FT1 had the lowest, while shrub biomass was similar in all fuel types.

### 3.2 Fire severity and combustion factors

The percentage of plots of green, scorch and charred severities varied among fuel types and species. Charred severity was predominant in oak forests (50-89%, Fig. 5a). In oak plots, FT3, with large trees but a second stratum and higher density than FT2, had the highest number of green plots (26 %), while FT1, with the lowest tree density, and FT4, with higher vertical and horizontal continuity and lower large trees, had the highest percentage of charred plots (89% and 72%, respectively, Fig. 5a). In pine forests, FT2, with large trees and low tree density, had the highest proportion of green plots (30%, Fig. 5b). FT2 and FT3 had the lowest percentage of charred plots (50%), while FT1 and FT4 had the highest number of charred plots (80 and 85%, respectively, Fig. 5b).

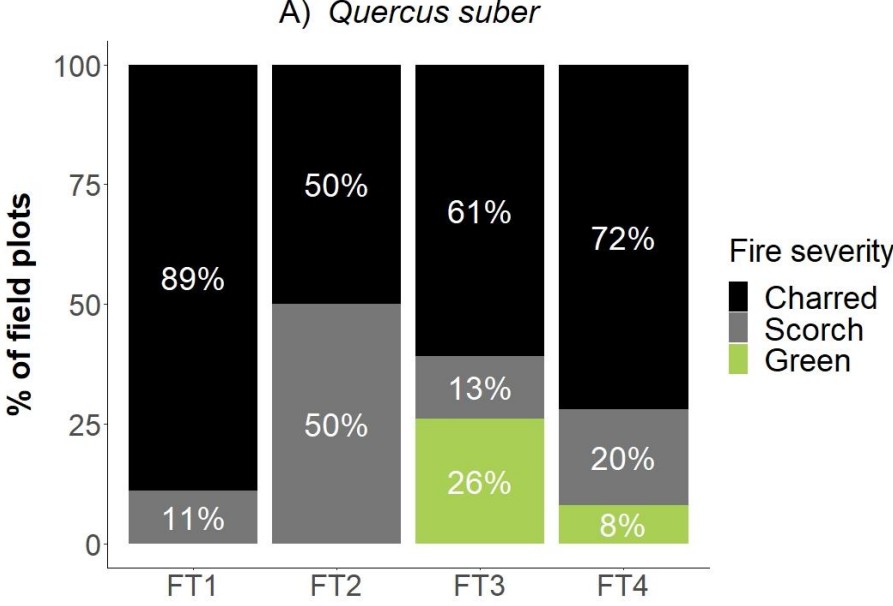



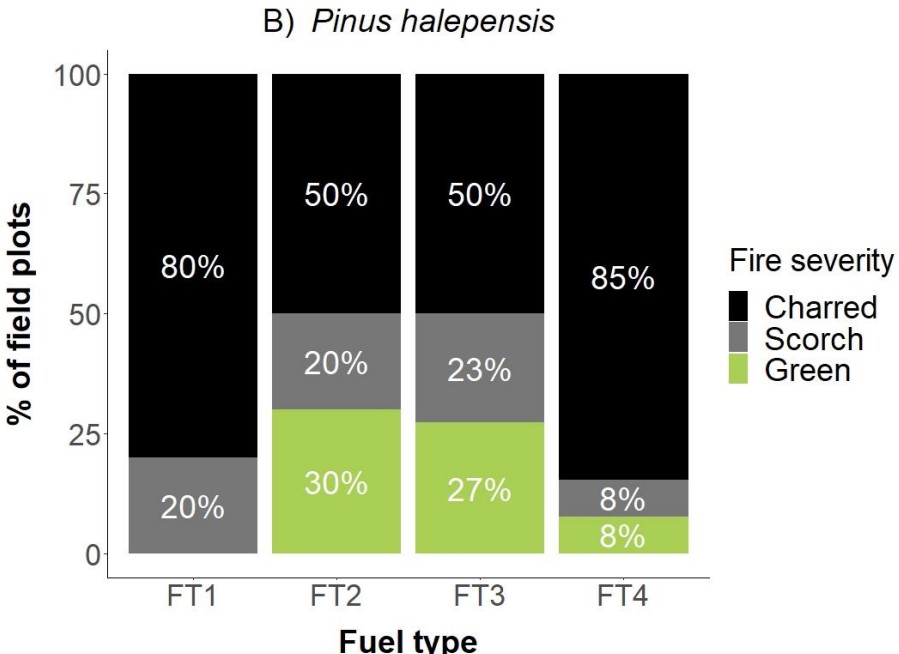

**Figure 5.** Percentage of plots with different fire severities (i.e., green, scorch, charred) and different fuel types (FT1 to FT4) of A) oaks, B) pines.

Combustion factors from the field were different among species and fuel types depending on fire severity (Table 1). In green plots, FT2 and FT3 of pines had the highest values of consumption of the litter and shrub layers and also, but to a lesser extent, of the crown fuels. FT3 and FT4 of oaks (the only available in the field) had lower consumption factors than their corresponding fuel types of pines (Table 1). In scorch plots, the two species had similar percentages of litter consumption, with higher values in FT1 and FT2. In the shrub layer, fine and coarse fuel consumption was higher in pine than in oak plots. Oak FT3 and FT4

had higher consumption factors, while they were similar in pine plots. Crown consumption factors were higher in FT1 than in the other fuel types for both species. In charred plots, litter and fine shrub available biomass were completely consumed and coarse shrub biomass was consumed from 90 to 98% in the different species and fuel types (Table 1). The coarse shrub fuels were consumed from 84 to 98% in the different fuel types and species. The percentage of crown leaves consumed in charred plots was near 100% in all cases, while a lower proportion of fine branches was consumed (67-82% in pines and 67-90% in

oaks).

**Table 1.** Combustion factors (% of available biomass combusted) for the different layers (crown leaves, crown fine branches, fine shrub fuel, coarse shrub fuel and litter), species (pines and oaks), fuel type (FT1 to FT4) and severity class (green, scorch and charred).

| Forest | Fuel type | Severity | Crown leaves | Crown fine branches | Fine shrub fuel | Coarse shrub fuel | Litter |
|--------|-----------|----------|--------------|---------------------|-----------------|-------------------|--------|
| Oak | FT1 | Scorch | 57 | 21 | 70 | 33 | 90 |
| | | Charred | 99 | 76 | 99 | 94 | 100 |





|  | FT2 | Scorch | 30 | 8 | 68 | 30 | 95 |
|---|---|---|---|---|---|---|---|
|  |  | Charred | 99 | 22 | 100 | 90 | 100 |
|  | FT3 | Green | 6 | 2 | 42 | 19 | 46 |
|  |  | Scorch | 38 | 22 | 97 | 82 | 85 |
|  |  | Charred | 100 | 82 | 100 | 95 | 100 |
|  | FT4 | Green | 7 | 2 | 42 | 20 | 38 |
|  |  | Scorch | 15 | 4 | 82 | 67 | 93 |
|  |  | Charred | 100 | 90 | 100 | 84 | 99 |
|  | FT1 | Scorch | 68 | 24 | 100 | 95 | 98 |
|  |  | Charred | 99 | 69 | 100 | 96 | 100 |
|  | FT2 | Green | 7 | 3 | 89 | 77 | 77 |
|  |  | Scorch | 20 | 11 | 100 | 95 | 99 |
|  |  | Charred | 96 | 67 | 100 | 98 | 100 |
| Pine | FT3 | Green | 15 | 6 | 69 | 48 | 62 |
|  |  | Scorch | 38 | 22 | 97 | 82 | 85 |
|  |  | Charred | 97 | 75 | 100 | 91 | 98 |
|  | FT4 | Green | 23 | 10 | 35 | 10 | 57 |
|  |  | Scorch | 25 | 13 | 95 | 70 | 90 |
|  |  | Charred | 98 | 82 | 99 | 90 | 99 |


### 3.3 Atmospheric pollutant emissions

We found that total emissions of all pollutants varied with fire severity, and $CO_2$ and $CH_4$ emissions varied with species (Table 2). In particular, $CO_2$, CO, $CH_4$ and $PM_{2.5}$ emissions were the highest in charred plots, followed by scorch plots and green plots, while $CO_2$ and $CH_4$ emissions were higher in pine than in oak plots (Fig. 6).


**Table 2.** Two-way ANOVA values of the effects of fire severity, species and their interaction on the emissions of $CO_2$, CO, $CH_4$, $PM_{2.5}$.

| Variable | Fire severity | Species | Fire severity*Species |
|---|---|---|---|
| $CO_2$ | **F=42.6, p<0.001** | **F=6.8, p=0.01** | F= 1.4, p=0.24 |
| CO | **F=21.6, p<0.001** | F=3.4, p=0.07 | F= 2.4, p=0.09 |
| $CH_4$ | **F=89.8, p<0.001** | **F=13.5, p<0.001** | F= 0.7, p=0.49 |
| $PM_{2.5}$ | **F=31.5, p<0.001** | F=2.9, p=0.09 | F=2.8, p=0.07 |





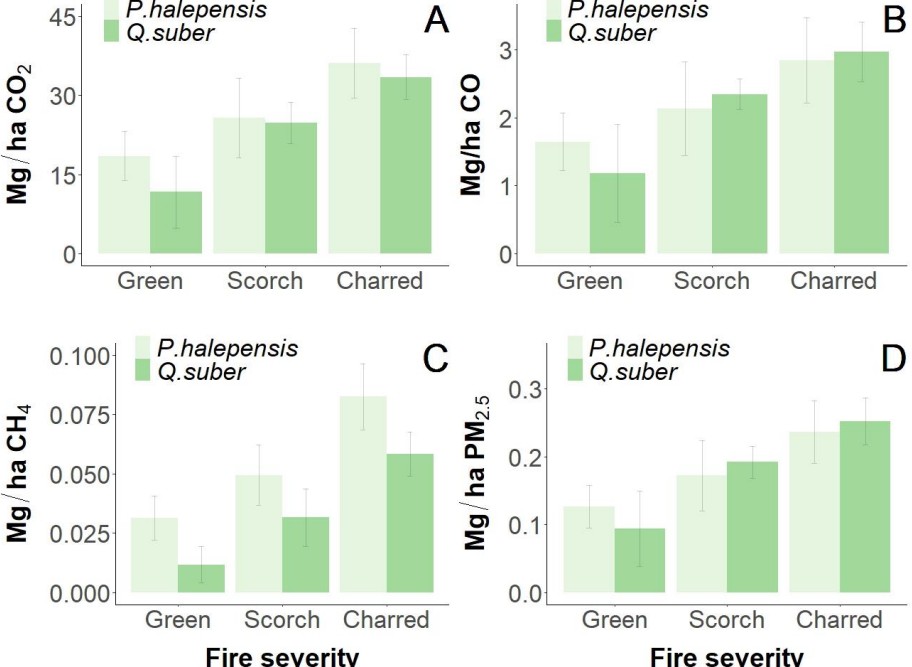


**Figure 6.** Mean (±standard deviation) values of A) $CO_2$, B) CO, C) $CH_4$, and D) $PM_{2.5}$ emissions (Mg/ha) in green, scorch and charred plots of pines (light green) and oaks (green).

Fire severity showed significant differences for all pollutants in the two forest types (Table 3), while there was no significant difference between fuel types, and the interaction of the two factors was only marginally significant in the case of $CH_4$ for 320 *Quercus* plots.

**Table 3.** Two-way ANOVA values of the effects of severity (green, scorch, charred), fuel type (FT2 and FT3 for pines, and FT 3 and FT4 for oaks) and their interaction on the emissions of $CO_2$, CO, $CH_4$, $PM_{2.5}$ of the two forest types considered separately. Significant differences among fire severities for each pollutant and species according to the Tukey post hoc test 325 are indicated with different letters.

**CO₂ emissions (Mg/ha)**

| Forest | Fuel Type | Severity | Fuel type * Severity | Green (mean±SD) | Scorch (mean±SD) | Charred (mean±SD) |
|---|---|---|---|---|---|---|
| Pine | F=0.6 p=0.46 | **F= 10.2 p<0.001** | F=0.0 p=0.99 | 18.9±4.8[B] | 25.1±7.9[B] | 38.6±6[A] |
| Oak | F=0.1 p=0.88 | **F= 43.0 p<0.001** | F=2.1 p=0.13 | 11.7±6.7[C] | 25.7±3.7[B] | 34.3±4.3[A] |

**CO emissions (Mg/ha)**



| Forest | Fuel Type | Severity | Fuel type * Severity | Green (mean±SD) | Scorch (mean±SD) | Charred (mean±SD) |
|--------|-----------|----------|----------------------|-----------------|------------------|-------------------|
| Pine | F=0.2 p=0.64 | **F=7.5 p=0.003** | F= 0.1 p=0.90 | 1.66±0.45[B] | 2.07±0.66[B] | 3.09±0.58[A] |
| Oak | F=0.2 p=0.64 | **F=27.2 p<0.001** | F=2.2 p=0.11 | 1.18±0.72[C] | 2.40±0.21[B] | 3.07±0.44[A] |

**$CH_4$ emissions (Mg/ha)**

| Forest | Fuel Type | Severity | Fuel type * Severity | Green (mean±SD) | Scorch (mean±SD) | Charred (mean±SD) |
|--------|-----------|----------|----------------------|-----------------|------------------|-------------------|
| Pine | F=1.1 p=0.3 | **F=21.5 p<0.001** | F= 0.1 p=0.90 | 0.033±0.008 [C] | 0.049±0.014 [B] | 0.087±0.01[A] |
| Oak | F=0.3 P=0.6 | **F=67.7 p<0.001** | **F=4.4 p=0.02** | 0.012±0.007 [C] | 0.034±0.012 [B] | 0.060±0.008[A] |

**$PM_{2.5}$ emissions (Mg/ha)**

| Forest | Fuel Type | Severity | Fuel type * Severity | Green (mean±SD) | Scorch (mean±SD) | Charred (mean±SD) |
|--------|-----------|----------|----------------------|-----------------|------------------|-------------------|
| Pine | F=0.4 p=0.5 | **F=9.0 p<0.001** | F=0.0 p=0.98 | 0.13±0.03 [B] | 0.17±0.05 [B] | 0.26±0.04[A] |
| Oak | F=0.1 p=0.7 | **F=35.4 p<0.001** | F=2.2 p=0.12 | 0.09±0.06[C] | 0.2±0.02[B] | 0.26±0.03[A] |


## 4. Discussion

### 4.1 Pre-fire available biomass

Total pre-fire available biomass is a key element to accurately estimating wildfire emissions. Although its estimation has improved with remote sensing methods, it has been based on crown biomass, often omitting considerable emissions from litter,
shrubs and young trees (Darío et al., 2018). We found small differences in available biomass (in crown, shrub and litter fuel layers) among oak and pine fuel types. The FT1 of pine stands, with higher open areas and low tree density, showed the lowest available biomass, while FT4, with higher vertical and horizontal continuity and lower large trees, reached the highest value in oak stands. In the absence of fire, available biomass was similar among fuel types and species, since vegetation grows until reaching its maximum potential disregarding differences among forest types. Our results are in line with previous analysis
showing similar amounts of available biomass (i.e.,10 to 30 Mg/ha for *Q. suber* and conifer forests) and distribution among layers, with a slightly higher percentage of litter in oaks but higher values of shrub and crown biomass in pines (Rosa et al., 2011). Litter represents the highest percentage of available biomass in oak stands, which agrees with the Ecological Forest Inventory of Catalonia data (average litter of 14.3 Mg/ha in *Q. suber* forests (Gracia, 2001)). In pine stands, the shrub layer contained the highest percentage of available biomass, followed by litter and crown (Fig. 4). Although our shrub biomass
estimates were below values found in other studies (i.e., 25-53 Mg/ha for pure maquis shrublands in Greece; Dimitrakopoulos, 2002), they are within the range of the understory in *P. halepensis* forests (Dimitrakopoulos et al., 2007). In this study, fine available crown biomass in pine stands was highly variable, from very low values in FT1 stands to the highest value of 9 Mg/ha in FT4 stands, this being below previously found values (Mitsopoulos and Dimitrakopoulos, 2007). Moreover, it is



common to include leaves and branches up to 0.6 cm when assessing crown biomass (Jiménez et al., 2013b), but we only
found branches below 0.6 cm consumed. For this reason, the percentage of biomass of the crown layer was lower than in other
works (Jiménez et al., 2013b; Balde et al., 2023).

### 4.2 Fire severity and combustion factors

In the Jonquera wildfire, we have found small differences in fire severity between oak and pine stands, with a dominance of
charred plots (66%) and a small presence of scorch and green severities (17 % of each) in the two species. However, fire
severity varied among fuel types. FT2, characterized by large trees and low tree density, was a rare fuel type in oak plots, but
it was common in pine plots and had the highest percentage of green plots (30%) (Fig. 5). Lower fire severity (in this case in
the two species) was found in FT3, a fuel type with similar forest structure to FT2 but with higher tree density and lower
percentage of large trees with a second tree layer. In pine plots, FT2 and FT3 showed lower fire severity than the other two,
with 50% of cases without charred plots (Fig. 5). These results are in line with those given by Alvarez et al. (2012a) in another
wind-driven wildfire from the same region. FT4 with higher densities and vertical continuity also had very high percentages
of charred plots in the two species. Oak forests burned by high-intensity fires is a phenomenon widely described in areas of
southern France near and north-eastern Catalonia (Schaffhauser et al., 2011; Sánchez-Pinillos et al., 2021). Finally, FT1 had
similar highest fire severity in stands of the two species, because it usually had the lowest tree density with open forest
structures, leading to higher wind speed and sometimes higher understory density, and all these characteristics increase the
availability of surface fuels to burn with higher intensity (Alvarez et al., 2012a; Sánchez-Pinillos et al., 2021).

In this study, combustion factors of the three layers (canopy, shrub and litter) (Table 1) were associated with a wind-driven
fire with high intensity surface fires and massive crown fires. Different researchers have related combustion factors to fire
behavior (surface or crown fire) (Molina et al., 2019), fire phase (smoldering or flaming), different biomes (van Leeuwen et
al., 2014) or other factors such as direction of forward spread of the fire (Surawski et al., 2016), wind speed or month of
occurrence (Fernandes et al., 2022). However, there is no data of the different combustion factors according to fire type
(topographic, convective or wind-driven fires). In la Jonquera fire, we found that litter, fine and coarse shrub biomass were
completely consumed (Table 1), and the consumption percentages followed the well-known relationship of higher fuel
consumption with higher fire severity (Molina et al., 2019; Price et al., 2022). *Pinus halepensis* and *Q. suber* forests have
highly flammable understory species such as *Rosmarinus officinalis*, *Quercus coccifera*, *Ulex parviflorus* or *Cistus spp*.
(Dehane et al., 2017; Sánchez-Pinillos et al., 2021; Pausas et al., 2012). Although crown leaves were completely consumed in
charred plots, something common in crown fires with high intensity (Mitsopoulos and Dimitrakopoulos, 2014), from 10 to 33
% of crown fine branches remained unconsumed. Wind-driven fires can spread with higher speed than convective or
topographic fires, resulting in partial branch combustion even when fire reaches the forest canopy (Jiménez et al., 2013b).
Interestingly, consumption was higher in pine than in oak plots in scorch and green severities. The low tree density but high
shrub fuel load in FT1 caused the highest fire severity and consumption in green and scorch plots. In pine plots, FT2 had the
highest proportion (>85%) of high trees (>8m) and lowest tree density (<1300 trees/ha), resulting in high litter and shrub
consumption but the lowest crown damage (Table 1). In this study this fuel type is the most fire-resistant, because it produced
high-intensity surface fires but low severity because of the presence of tall trees. Similarly, Busby et al. (2023) also found that
overstory canopy height was the only influential forest structure variable in reducing fire severity under periods of high winds.

### 4.3 Atmospheric pollutant emissions



Quantifying and comparing wildfire emissions is challenging because their amount and chemical composition vary greatly among fires (Sommers et al., 2014). In our study, fire severity was the main factor that determined emissions for all pollutants. It is well known that fire severity determines burning efficiency and the level of emissions (Balde et al., 2023; De Santis et al., 2010). Emissions of $CO_2$ were slightly higher in pine than in oak forests. The Jonquera fire emitted 34 to 39 Mg/ha of $CO_2$ in charred plots and 12 to 19 Mg/ha in green plots of oak and pine, respectively (Table 3). These values are far lower than the $CO_2$ emissions of a 100,000 ha fire in Australia (Surawski et al., 2016), but are in line with studies in the same region (Marino et al., 2017; Chaves Naharro, 2015; Valero et al., 2007) and climate (Bacciu et al., 2009), with some variations depending on the forest type and severity. However, these studies did not consider litter and included crown branches higher than 0.6 cm (which we did not find consumed). Although the methodology to estimate crown biomass consumption is continuously improving, wildfire consumption remains underestimated because litter and shrub fuel layers are often not included (Domingo et al., 2017).

Epidemiological literature has focused on the impact of concentrations of $PM_{2.5}$ on human health but estimating CO emissions is gaining importance because CO increases during fires also adversely affect the life of all breathing creatures until one month after the fire (Griffin et al., 2023; Yilmaz et al., 2023). Here, we only assessed immediate CO and $PM_{2.5}$ emissions, with lower values of CO (5.3 Mg/ha) and $PM_{2.5}$ (0.5 Mg/ha) than in the Andilla fire but similar to the maquia forest fire of Cortes de Pallás (Chaves Naharro, 2015). In contrast, Bacciu et al. (2009) gave values from 0.3 to 7 of CO and from 0.04 to 0.08 of $PM_{2.5}$ in a Mediterranean maquis fire. For $CH_4$ emissions, there is little information in Mediterranean wildfires due to the lack of data on emission factors compared to other countries such as the USA (Urbanski et al., 2022; Prichard et al., 2020). We found higher $CH_4$ emissions in pines than in oaks (Table 2 and 3). Moreover, in pines we observed higher differences in emissions of $CH_4$ among fire severities than for the other gases. Pines have greater amount of resin than oaks, and this compound is rich in carbon and could favor the emission of $CH_4$. The $CH_4$ values of this study were higher than emissions in Andilla and Cortes del Pallás (0.037 and 0.022 Mg/ha) and also much higher than those of Bacciu et al. (2009), which were 0.019-0.036 Mg/ha. Moreover, the higher consumption rates in pines than in oaks at low fire severity could compensate for the higher litter biomass found in oak forests. On the other hand, shrubs associated with oak forests have greater water retention capacity and greater resistance to fire than those of pine stands, which could imply less combustion and less $CH_4$ emissions.

## 5. Conclusions

Unprecedented wildfires are affecting all ecosystems around the globe and are raising the emissions of gases (Shakoor et al., 2023). Remote sensing tools are important to estimate emissions in wildfires, but field data is needed to improve and validate pre-fire available biomass and consumption factors in different fire types (Jiménez et al., 2013b; Ottmar, 2014; Domingo et al., 2017). We found that pre-fire available biomass was distributed differently in *Q. suber* and *P. halepensis* stands, with a higher percentage of litter than shrub biomass in the former and more crown biomass in the latter. Future studies should complement crown fuel measures with estimates of the litter and shrub layers, because these surface and ground fuels are the most strongly consumed in a wildfire. The effect of fire severity on the different fuel types was also different between species. In pine stands, FT2, with large trees and low tree density, was the fuel type with the highest proportion of green severity (Fig. 5), but it was very rare in oak forests. In oaks stands, FT3, with lower tree density and vertical continuity than FT4, had more cases with no charred severity. Pine stands have more available biomass in the shrub layer than oak, and they also have taller trees that can better withstand surface fires. This could explain the differences in $CO_2$ and $CH_4$ emissions and the greater resistance of pines to low fire severities. As wildfires continue to increase, estimating pollutant emissions from shrub and litter



would definitely improve the overall accuracy of wildfire emissions to better support forest policy and management in effectively addressing the increase of pollutant emissions to the atmosphere.

*Data availability.* Data will be made available on request.

*Author contribution.* **Albert Alvarez:** Conceptualization, Data curation, Formal analysis, Investigation, Methodology, Project administration, Software, Supervision, Visualization, Writing - original draft, Writing - review & editing **Judit Lecina-Diaz:** Conceptualization, Data curation, Investigation, Software, Writing - review & editing **Miquel De Cáceres** Data curation,
Software, Writing - review & editing **Jordi Vayreda** Conceptualization, Data curation, Investigation, Methodology, Software, Writing - review & editing **Javier Retana:** Conceptualization, Formal analysis, Methodology, Project administration, Supervision, Visualization , Writing - original draft ,Writing - review & editing.

*Competing interests.* The authors declare that they have no known competing financial interests or personal relationships that
could have appeared to influence the work reported in this paper.

*Acknowledgements.* This work has been funded by the project GREEN-RISK "Evaluation of past changes in ecosystem services and biodiversity in forests and restoration priorities under global change impacts" (PID2020-119933RB-C21/AEI/10.13039/501100011033), funded by the Spanish Ministry of Science and Innovation. We thank The Generalitat de Catalunya fire-fighters corps for providing the helicopter flights as well as the use of the Llançà fire station during field work
and the GRAF unit for some reports and information gathered from visits to the fire. Thanks to Paul Abbott for English language consultation.

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
