# Peer review of "Fuel types and fire severity effects on atmospheric pollutant emissions in an extreme wind-driven wildfire"

_EGUsphere, 2024_

## Author Comment (AC1)

**Answer to Reviewer #1: Anonymous Referee**

**Manuscript presents and interesting exercise of evaluation of consumed biomass and emissions estimated during a wildfire and the differences between Pinus and Quercus stands.**

Thank you very much for your reply. We appreciate the time the reviewer spent on reviewing the manuscript. We have considered all of the reviewer comments and suggestions, and either incorporated them into the text or responded to them below.

**The exploitation of public databases and field sampling is a valuable approximation to this topic. Nevertheless, this method present important limitations that must be highlighted by authors. A high level of uncertainty is expected with proposed methodology; therefore authors and readers must be aware of this approximation to results and derived conclusions.**

We agree with your comment about the important uncertainties of the method in each one of their steps. We have written a short section in the discussion with the different uncertainties from the method and what we tried to do to reduce these uncertainties **"Section 4.4. Uncertainties in emissions estimates and limitations" on (Lines X).**

We used Seiler and Crutzen (1980) method and the equation;

EM = A x B x C x D,

where EM are the total emissions (Mg/ha), A is the area burned (ha), B is the available biomass before the fire (Mg/ha), C is the combustion factor (%) and D is the emission factor (g/kg).

If the burned area is precisely defined, as we think it was in our case, the greatest uncertainties are in the estimation of biomass prior to the fire and the combustion factor (Ottmar et al., 2008; Bacciu et al., 2015; Fernandes et al., 2022), but also on the use of emission factors available. We discuss below each of these components.

The quantification of different fuel types before the fire is one of the main sources of uncertainty, indeed, variations in fuel characteristics **may contribute to 83 percent** uncertainties in estimates of wildfires emissions (Ottmar et al., 2008; Fernandes et al., 2022). Moreover, in the south-European forests, the high spatial variability of fuel loadings, the fuel structure which describes how fuel loads are vertically distributed in a stand and how fire moves through a site are critical factors when describing forest fuels and determine final pollutant emissions (Carvalho et al., 2007; French et al., 2011). **We have tried to reduce this uncertainly** with the identification of species at different layers and by using specific allometric equations from a combination of data from National Forest Inventory (NFI) for canopies and shrubs but especially from Ecological Forestry Inventory of Catalunya for litter assessment (IEFC) (Gracia, 2001; Vayreda et al., 2016). Although biomass allometries are also a source of uncertainty, the range values in which they move fit with real data from the inventories and are at least differentiated between plots instead of using a general value for all plots. There are more precise allometric equations for shrublands in Spain that we used but they are also difficult to apply after the fire when sometimes you cannot identify variables such as height or diameter of the shrub. The joint of these data allowed reduction in the uncertainty of the total fuel load available **especially including the litter** component that is not usually assessed.

In fact, previous studies considered that forest floor emissions were the most uncertain component when modelling carbon emissions from forest fires, since its consumption can range from near 0

to 100% (Vilen and Fernandes 2011). In addition, considering the difficulty in estimating combustion of subsurface carbon and that 65% of the total fire-wide carbon emissions may come from the combustion of litter, duff, and mineral soil carbon (Campbell et al., 2007), we consider that most of the uncertainty in our estimate of total emissions may arise from uncertainty in combustion of these fuels. Therefore, our results should be taken with caution because of the complexity of litter distribution and the variables that could influence its amount and variation. We highly recommend to develop better models for assessing litter for different forest species because of its high percentage in the two forest types studied. This will be the future challenge since for the moment remote sensing is not useful for assessing these fuel loads. If the content of litter is not assessed, there will be a constant underprediction of fuel load and emissions especially when there are extreme wildfires.

**Combustion factors** for each layer are another important source of uncertainty with values over 30% (Ottmar et al., 2008; Fernandes et al., 2022) and values that could be higher depending on the position of the fire (head, flank, back) (Surawski et al., 2016). There are few examples from field works especially when you apply Seiler and Crutzen (1980) method with remote sensing tools (De Santis et al., 2010; Jimenez et al., 2013). However, we reduced that uncertainty with direct observation of the percentage consumed after the fire at the three layers. This is especially important when you are comparing emissions between different fire severities. It is also important to distinguish what crown fraction is consumed in crown fires in wind fires since it is usually overestimated because it is often considered that up to 0.6-2.5cm all branches are consumed when, in this fire type, we saw that in some cases not all the fine material had been consumed. Although the estimation method was visual, it helps to understand what fraction of the canopies can be consumed in this type of fires. However, it would be necessary to replicate the measures in convective and topographic fires where perhaps more canopies can be consumed. Overall, these variations in crown consumption could affect the total biomass but the largest amount of biomass consumed is clearly that of shrubs and litter, the biomass most poorly estimated in studies on emissions and therefore the greatest source of uncertainty.

**Regarding emission factors,** they are also one of the main uncertainty sources in emissions estimations. It is important to work on this topic with more field measurements, in particular for southern European conditions variability (Fernandes et al. 2022). EFs variation (mainly due to type of pollutant, type and arrangement of fuel, and combustion factor) that could **contribute to about 16% of the total error** associated with emissions (Ottmar et al., 2008; Bacciu et al., 2015; Fernandes et al., 2022) is mainly available for United States of America (USA) forests (Urbanski, 2013), but it is not a suitable proxy for wildfires in Europe, due to the different vegetation cover and the differences in combustion characteristics (e.g. flaming and smouldering phases).

There are also other limitations with the methods used in the current study. They include not considering fuel load from herbs, not taking into account the influence of topography or fuel moisture on the general emission factors and the use of litter emission factors from *Q. pubescens* (the only available) instead of those from *Q. suber*. We also did not differentiate between flaming or smoldering phase of combustion and we did not consider fuel moisture.

References:

Bacciu, V., Spano, D., and Salis, M.: Emissions from Forest Fires: Methods of Estimation and National Results, in: The Greenhouse Gas Balance of Italy: An Insight on Managed and Natural Terrestrial Ecosystems, edited by: Valentini, R. and Miglietta, F., Springer, Berlin, Heidelberg, 87–102, Germany, https://doi.org/10.1007/978-3-642-32424-6_6, 2015.

Campbell, J., Donato, D., Azuma, D., and Law, B.: Pyrogenic carbon emission from a large wildfire in Oregon, United States, J. Geophys. Res. Biogeosciences, 112, https://doi.org/10.1029/2007JG000451, 2007.

Carvalho, A., Monteiro, A., Flannigan, M., Solman, S., Miranda, A., and Borrego, C.: Forest fire emissions under climate change: impacts on air quality, in: Seventh Symposium on Fire and Forest Meteorology, The Turrets, USA, 23 October 2007, https://ams.confex.com/ams/7firenortheast/techprogram/paper_126854.htm (last access: 23 April 2024), 2007.

De Santis, A., Asner, G. P., Vaughan, P. J., and Knapp, D. E.: Mapping burn severity and burning efficiency in California using simulation models and Landsat imagery, Remote Sens. Environ., 114, 1535–1545, https://doi.org/10.1016/j.rse.2010.02.008, 2010.

Fernandes, A. P., Lopes, D., Sorte, S., Monteiro, A., Gama, C., Reis, J., Menezes, I., Osswald, T., Borrego, C., Almeida, M., Ribeiro, L. M., Viegas, D. X., and Miranda, A. I.: Smoke emissions from the extreme wildfire events in central Portugal in October 2017, Int. J. Wildl. Fire, 31,  989-1001, https://doi.org/10.1071/WF21097, 2022.

French, N. H. F., de Groot, W. J., Jenkins, L. K., Rogers, B. M., Alvarado, E., Amiro, B., de Jong, B., Goetz, S., Hoy, E., Hyer, E., Keane, R., Law, B. E., McKenzie, D., McNulty, S. G., Ottmar, R., Perez-Salicrup, D. R., Randerson, J., Robertson, K. M., and Turetsky, M.: Model comparisons for estimating carbon emissions from North American wildland fire, J. Geophys. Res., 116, https://doi.org/10.1029/2010JG001469, 2011.

Jiménez, E., Vega, J. A., Ruiz-González, A. D., Guijarro, M., Varez-González, J. G., Madrigal, J., Cuiñas, P., Hernando, C., and Fernández-Alonso, J. M.: Carbon emissions and vertical pattern of canopy fuel consumption in three Pinus pinaster Ait. active crown fires in Galicia (NW Spain), Ecol. Eng., 54, 202–209, https://doi.org/10.1016/j.ecoleng.2013.01.039, 2013.

Ottmar, R. D., Miranda, A. I., and Sandberg, D. V: Chapter 3 Characterizing Sources of Emissions from Wildland Fires, in: Developments in Environmental Science Wildland Fires and Air Pollution, vol. 8, edited by: Bytnerowicz, A., Arbaugh, M. J., Riebau, A. R., and Andersen, C., Elsevier, 61–78, 2008.

Surawski, N. C., Sullivan, A. L., Roxburgh, S. H., and Polglase, P. J.: Estimates of greenhouse gas and black carbon emissions from a major Australian wildfire with high spatiotemporal resolution, J. Geophys. Res. Atmos., 121, 9892–9907, https://doi.org/10.1002/2016JD025087, 2016.

Urbanski, S. P.: Combustion efficiency and emission factors for wildfire-season fires in mixed conifer forests of the northern Rocky Mountains, US, Atmos.Chem.Phys., 13, 7241–7262, 2013.

Vilen, T. and Fernandes, P. M.: Forest fires in Mediterranean countries: CO2 emissions and mitigation possibilities through prescribed burning, Environ. Manage., 48, 558–567, 2011.

**Next, I detail some comments and suggestions to improve the manuscript and some question to authors that must be clarified:**

**Line 80. The main objective is not completely in agreement with title. Reconsider rewrite de title please. Propose a hypothesis please.**

OK, we will reconsider both the title and the formulation of the main objective to make them coherent between them. However, as the other reviewers may also propose changes in this direction, we prefer to wait for having all comments and then reformulate them.

**Line 102. Regeneration of Quercus ilex under Q. ilex stands?**

No, this was incorrectly written it was *Quercus suber* under *Q. suber* stands. We have changed that on **line 102.**

**Line 102**. **What about mixed forest? Are there mixed oak-pine forests in burned area?**

Yes, there were areas with mixed forests. In the north-west forests of *Q. suber* changed to *Q. ilex*, but this was burned with south winds in the days following the start of the fire and there were not enough cases to include in the study. There were also mixed areas of *P. halepensis* and *Q. ilex*, but this type of forest was not dominant in the landscape and we only carried out plots in areas with pure forest species, either pines or oaks.

**Line 112. This is not considering a large spot distance during extreme wildfire events (e.g. see Tedim et al. 2019)**

You are right, spot distances of 200-400 meters with a maximum of 1km are not considered large spot distances but intermediate-range spotting during this extreme wildfire event EWE (Tedim et al., 2018). This is especially true when you compare with other countries and forest types (Martin and Hillen, 2016; Cruz et al., 2012) and accepted definitions to describe extreme fire event (Tedim et al., 2018).

We have upgraded the section 2.2 "Fire description and weather conditions during the fire" to justify better the reason because this fire was considered an EWE following Tedim et al., (2018) among other articles.

References:

Cruz, M. G., Sullivan, A. L., Gould, J. S., Sims, N. C., Bannister, A. J., Hollis, J. J., and Hurley, R. J.: Anatomy of a catastrophic wildfire: the Black Saturday Kilmore East fire in Victoria, Australia, For. Ecol. Manage., 284, 269–285, 2012.

Bombers. Report of the Jonquera wildfire. Bombers de la Generalitat de Catalunya, Departament de interior de la Generalitat de Catalunya. https://agricultura.gencat.cat/web/.content/06-medi-natural/boscos/gestio-forestal/obres/restauracio-forestal/restauracio-hidrologica/fitxers-binaris/jonquera_informe_incendi.pdf (last access: 23 July 2024), 2012.

DARP: Report on the forest fire of July 22, 2012 in La Jonquera (Alt Empordà), Generalitat de Catalunya, Departament d'Agricultura, Ramaderia, Pesca, Alimentació i Medi Natural, Girona, Spain, https://agricultura.gencat.cat/web/.content/06-medi-natural/boscos/gestio-forestal/obres/restauracio-forestal/restauracio-hidrologica/fitxers-binaris/jonquera_informe_incendi.pdf (last access: 23 July 2024), 2012.

Martin, J. and Hillen, T.: The spotting distribution of wild fires, Appl. Sci., 6, https://doi.org/10.3390/app6060177, 2016.

Tedim, F., Leone, V., Amraoui, M., Bouillon, C., Coughlan, R. M., Delogu, M. G., Fernandes, M. P., Ferreira, C., McCaffrey, S., McGee, K. T., Parente, J., Paton, D., Pereira, G. M., Ribeiro, M. L., Viegas, X. D., and Xanthopoulos, G.: Defining Extreme Wildfire Events: Difficulties, Challenges, and Impacts, Fire 2018, 1(1), 9; https//doi.org/10.3390/fire1010009, 1, 2018.

**Line 120. "monspeliensis" Lowercase**

OK, changed (Line 120).

**Line 125.** **Why do you not use remote sensing data to plan the inventory? Helicopter flight do not seem a very economic method and you probably obtain similar categories than remote sensing from Copernicus database (dNBR). Justify better please.**

After the fire, during August 2012 we did not know if we would have the opportunity to do the study and some of us were working with firefighters collecting data from fires. Until September 2012 we could not start to plan the field work and we did not have much time because we knew that the timber extraction work was going to start immediately as it was happened in other fires where we did field work (Alvarez et al., 2012).

A recording flight of the fire had to be carried out by the fire department, so due to our collaboration with them, we were able to participate in the flight, obtaining videos and hundreds of photos of the entire fire. Therefore, it was not a specific flight for the work but a complementary activity. These were used to make an initial estimate of the burned area by each severity and to locate areas where to find severities that were not charred. In fact, we thought about the possibility of apply any remote sensing index but we did not know if it could identify these areas with lower severity better than with the flight material obtained. Finally, we were lucky to participate in the flight because we could also understand better the global spread of the fire that helped us to obtain other data related to fire behavior.

In section 2.3 "Field plot data and fire severity estimation" we have included a sentence to clarify the reason because we used a flight instead of the use of remote sensing tools to determine distribution of fire severity. We will highlight the lack of time, the opportunity of obtaining a multifunctional data from flights and the doubt about the precision to distinguish green and scorch areas in this massive scorch wildfire.

References:

Alvarez, A., Gracia, M., Castellnou, M., and Retana, J.: Variables That Influence Changes in Fire Severity and Their Relationship with Changes Between Surface and Crown Fires in a Wind-Driven Wildfire, For. Sci., 59, 139–150, https://doi.org/10.5849/forsci.10-140, 2013.

**Line 150-155.** **This hierarchical and deterministic classification following Alvarez et al. 2012 must be justified for studied area. e.g. using cluster analysis**

We used this classification in the studied area because it was created and tested in a fire only 30 km away from this fire in Pinus halepensis stands, one of the two main species of forest types of this study with also similar fuel types. Moreover, the classification was made not only to apply to one species but in general, according to common but critical variables that determine fire behavior independently of forest types: the density of trees, which determines horizontal continuity, and the percentage of trees based on height, which determines vertical continuity.

At the beginning of the study, we concluded that we captured the variable forest structures in Q. suber plots, just as the classification method was designed to be applied to other types of forests and allow comparisons. The result was really good because it was possible to identify the differences in the fuel types between the two different types of forests and the effects on fire severity and consumption. We have added a small comment justifying this point in section "2.3 Field plot data and fire severity estimation".

**Line 170-174. How? Visually estimation?** **In my opinion it is very difficult this estimation at crown level and the uncertainty of measure is very high. Consumed visually observed could be a good estimation if pre-fire data are available (very difficult or impossible during**

**wildfires). In my opinion a good estimation of percentage of shrub and dead fuel data in opportunistic sampling need a comparison between burnt and unburnt plots (control unburnt plots is needed to ratify data obtained in 3FNI see below).**

We agree with you about the difficulty of the visual estimation of fine crown and shrub percentage after the fire and we also understand the concern about the high degree of uncertainty that this measure can cause. However, according to De Santis et al. (2010), biomass consumption was traditionally estimated using a two-step methodology which includes:

a) the estimation of pre-fire biomass by applying allometric regression equations using destructive sampling or biomass values per species and
b) the post-fire biomass estimated by field-based weighting or by visual examination.

When we started the fieldwork, we visited the few areas unburned within the fire perimeter to understand what possible fuels we could find and to identify species. Moreover, we visited areas that immediately bordered the perimeter of the fire when we had plots near the perimeter. When we started to measure the plots, we invested long time to measure all the different possible diameters from shrubs and fine branches from trees with a caliper.

All plots were done by the same two people in order to avoid observer bias that could cause a significant influence in the kind of measures that we took, the percentages of fuel types after the fire. The value of each percentage was an average value from the two people to avoid errors of perception. We also take dozens of photos from all angles from each plot to capture trees, shrubs and litter. It was useful because at the beginning every night we contrasted percentages given to each plot and adjusted them comparing with previous plots when it was necessary. After this first training and as we made more plots, we had a more balanced vision (which was far from perfect) of the percentages we gave, so that the quantitative differences we appreciated were relatively small. The fieldwork was intensive from Octobre 2012 to March 2013, only stopped on very windy or rainy days. At the end of the work, we obtained near 12,000 photos from plots and their surrounding area that also helped us to calibrate dubious plots.

On the other hand, we transparently recognize that the potential shrub or litter cover measurements before the fire based on the number of shrubs, and comparing it with what we saw in unburned areas inside could be not as accurate as it could be using other methods. However, these quantitative percentages reflected the difference between plots that we saw qualitatively and the results obtained from the fuel loads from shrubs were within the ranges that we obtained from the IFN3 plots and bibliography.

We **have updated the section "2.4.1 Area burned and pre-fire available biomass",** including a synthesis of the training method to obtain the percentage of fuel consumed. Moreover, in the discussion or/and in the new **section "4.4 Uncertainties in emissions estimates and limitations"** we have included the implications over the uncertainties and potential overestimation of shrub and crown fuel loads before and after the fire.

References:

Ottmar, R. D., Vihnanek, R. E., and Wright, C. S.: Stereo photo series for quantifying natural fuels Volume X : sagebrush with grass and ponderosa pine-juniper types in central Montana, USDA For. Serv. Pacific Northwest Res. Station. Gen. Tech. Rep., X, 2007.

De Santis, A., Asner, G. P., Vaughan, P. J., and Knapp, D. E.: Mapping burn severity and burning efficiency in California using simulation models and Landsat imagery, Remote Sens. Environ., 114, 1535–1545, https://doi.org/10.1016/j.rse.2010.02.008, 2010.

Surawski, N. C., Sullivan, A. L., Roxburgh, S. H., and Polglase, P. J.: Estimates of greenhouse gas and black carbon emissions from a major Australian wildfire with high spatiotemporal

resolution, J. Geophys. Res. Atmos., 121, 9892–9907, https://doi.org/10.1002/2016JD025087, 2016.

**Line 197**. **To my understanding NFI data from shrubs are a estimation of 5 m radius plot in the centre of NFI plot, is it correct? Authors are expanding these data to plot level in field data. Be cautious please.**

**At least authors must be honest highlighting the limitation of these data. In addition, what is the time lag between NFI3 and wildfire? Authors must be highlight or justify how including the growing of shrubs on results. On the contrary they must assume the underestimation of biomass during the combustion process. It is important because this value affects to estimated emissions.**

**I recommend consulting models proposed by Montero et al. for potential estimation of shrub growing. This work includes correlation models for all Shrub communities in Spain and could be useful to compare results with models proposed by authors**.

For the first point, NFI data from shrubs are a measure of 10 m radius plot in the center of each NFI plot (https://www.mapa.gob.es/es/desarrollo-rural/publicaciones/publicaciones-de-desarrollo-rural/librobiomasadigital_tcm30-538563.pdf). Regarding the use of IFN3 plots in shrub load assessment, we developed models that related tree cover (at the time of IFN3) with shrub fuel (also at IFN3), estimated with the model MEDFATE (De Càceres et al., 2019). Then these allometries are applied to the field data. Field data for shrubs were the species identification and the shrub fraction cover, which was used in the equations created with the data from the IFN3 results applying MEDFATE. We have included in the section "4.4 Uncertainties in emissions estimates and limitations" the potential overestimation of biomass with the measure of shrub cover. However, after the fieldwork and comparing plots this percentage was logical with the qualitative estimation among all the different plots.

IFN3 plots were done from 2000 to 2001 and the fire was in 2012. However, we did not used the original values of those plots ten years later. We used IFN3 plots combined with measures of MEDFATE model to obtain allometric equations used later with the field data to assess the shrub fuel load. Therefore, there was not a problem with the growing of shrubs in the IFN plots.

References:

De Càceres, M., Casals, P., Gabriel, E., and Castro, X.: Scaling-up individual-level allometric equations to predict stand-level fuel loading in Mediterranean shrublands, Ann. For. Sci., 76, 87, https://doi.org/10.1007/s13595-019-0873-4, 2019
Pasalodos-Tato, M., Ruiz-Peinado, R., Ri¢, M. de., and Montero, G.: Shrub biomass accumulation and growth rate models to quantify carbon stocks and fluxes for the Mediterranean region, Eur. J. For. Res., 134, 537–553, 2015.

**Line 215**. **Two months after fire (date of sampling) most of scorched needles in moderate and low severity fire have fallen. Explain better how you estimate % of consumed biomass please. In my opinion a visually observed evaluation must be carried out 1-2 week after fire in order to classify completely burned, scorched and not consumed crown fuels. Explain this point better please, or assume the limitation from this estimation**

The question about how we estimated the % of consumed biomass has been explained in detail in the previous question "Line 170-174.".

Regarding the fact that needles fall before measuring the amount remaining on the tree after the fire, it is an uncertainty that is inevitable. It is true that as more time passes after the fire, the number fallen needles in areas of moderate/low severities increases. We took this phenomenon into account by observing the number of needles on the burned ground, which were therefore not present previously. We considered whether there was a slope to determine if the needles in the area could correspond to the sampled trees and the condition of the trunk and branches of the trees before decide the percentage of needles unburned.

Moreover, the fall of the needles, and in general the alteration of the conditions of the plots, depends greatly on the days of rain and wind that occur after the fire. In 2012, after the fire, the number of rainy days in the area burned was very few. Despite some windy days, the interruptions to fieldwork were generally very few (perhaps less than five). We have also included these considerations warning of the effect on the measurements of this phenomenon, and that it was taken into account in section "2.3 Field plot data and fire severity estimation".

**Line 225**. **I aware the difficulties to obtain emission factors for all species studied but in my opinion Q. pubescens is a very different species and ecosystem than Q. suber**

Yes, it is also true that *Q. pubescens* and *Q. suber* ecosystems may be quite different. However we only used the emission factors for the litter component from Pallozi et al. (2018) because it was the only available study and because at least *Pinus halepensis* was correctly represented and the other species was of the same genus. We also have included this consideration in the section "4.4 Uncertainties in emissions estimates and limitations".

References:

Pallozzi, E., Lusini, I., Cherubini, L., Hajiaghayeva, R. A., Ciccioli, P., and Calfapietra, C.: Differences between a deciduous and a conifer tree species in gaseous and particulate emissions from biomass burning, Environ. Pollut., 234, 457–467, https://doi.org/10.1016/j.envpol.2017.11.080, 2018.

**Line 226**. **ANOVA assumes independent and randomized events for each plot. This is not true in studied plots (Figure 1). I suggest including a spatial correlation analysis**

We will try to test the presence of spatial autocorrelation in the new version of the manuscript.

**In my opinion results and discussion could be different if method is refocused. Authors must justify well their decision to assess robust results**

We agree with this point and we have included a new paragraph that we hope could reflect the uncertainty and limitations of the work done and its potential implications. After your comments we expect to explain better the weakest points of the work, recognizing the limitations but highlighting the potential knowledge it can also provide.

---

## Author Comment (AC3)

**Reviewer #3: Anonymous Referee**

**Thanks to their authors for their submission. The fire science discipline is always advantaged by studies on pre-burn biomass determination followed by assessing the impact of fire behaviour on fuel consumption and emissions. In a changing climate, such investigations are worthwhile so well done on getting the work to this point. Some corrections are required before a favourable decision can be reached on this article. Two higher-level questions involve:**

Thank you very much for your reply. We appreciate the time the reviewer spent on reviewing the manuscript. We have considered all of the reviewer comments and suggestions, and either incorporated them into the text or responded to them below.

**Statistical analysis. Multiple linear regression and two/three-way Analysis of Variance is used in this manuscript. I would recommend that the authors check (and report upon) whether the assumptions underpinning these techniques are satisfied or not. The conclusions of this manuscript hinge on statistical analysis of results so a robust effort is required here.**

What we have done with all parametric tests is to represent the residuals of the models with the predicted values. Examining the graphical representation of the residuals against the expected values allows us to assess a series of assumptions made about the quality of the model fit: (i) Normality: the residuals are assumed to be normally distributed around each predicted value; (ii) Linearity: it is also assumed that there is a linear relationship between the residuals and the predicted values; (iii) Homoscedasticity: it is also assumed that the variance of the residuals is similar for different values of the dependent variable.

If the editor considers it appropriate to show all residual graphics in the supplementary material, we will include them.

**The discussion section needs more work. In my opinion two extra sections are required. 1) An additional sub-section would compare your results with other inventories in your country either at the regional or national level. This will make it easier for the reader to see how your estimates quantitatively compare with previous work. 2) A section should be added on uncertainties in your emissions estimates since you rely on information source with error e.g. allometric equations for biomass determination, plot level sampling errors and emission factors with uncertainties.**

We agree with your two suggestions. First, we have upgraded the discussion including more references to compare our results with other works or methodologies used to measure wildfire emissions. Second, there is a new section "4.4 Uncertainties in emissions estimates and limitations", which includes the various limitations and uncertainties at the different levels of emissions estimation (field work, different components of the calculation method, i.e. calculation of the pre-fire fuel load, estimation of the combustion factor, emission factors, etc.). We have included in this section all comments from all of reviewers to enrich and clarify how can be considered the results of this study.

**Some other suggested corrections are:**

**The phrase 'wind-driven wildfire' is used in the manuscript. Is there such a thing as 'non-**

**wind-driven wildfire'. I thought wind would always be a necessary component for wildland fire.**

This terminology started to be used more frequently after the Fire Paradox project in Europe (Silva et al, 2010). The European Project "Fire Paradox" analyzed the spread of fire in historical wildfires and showed that there were similar spread schemes dominated by common factors (e.g. wind direction and speed). Depending on the spread scheme and the dominant spread factor, three fire types were defined: convection or plume dominated fires, wind-driven fires and topographic fires (Castellnou et al., 2013; Costa et al., 2011). Firstly, convection or plume-dominated fires are characterized by the accumulation of high quantity of available fuels and atmospheric instability. This fire type has such a high intensity and extreme behavior that produces its own fire environment and generates massive spotting. Secondly, wind-driven fires follow the speed and direction of strong winds when the meteorological window that produces the fire conditions is maintained, with the same intensity and velocity during day and night. In both of them, small changes in the landscape have little influence in the direction and behavior of these fire types, especially under extreme meteorological conditions. In contrast, topographic fires are dominated by local winds caused by slope and differences in solar heating of the earth surface (i.e. sea breeze, land breeze, valley and slope winds). The direction of this fire type changes with topography (e.g. hydrographic basins, main valley), and it has high intensity during the day and low intensity at night (Castellnou et al., 2013; Costa et al., 2011). In the latter fire type, wildfire is more sensitive to small changes, thus little variations of topographical wind, slope or aspect have higher influence on fire behavior (Lecina-Diaz et al., 2014).

The combination of two or three fire types in the same wildfire might be common in North America, Canada and Australia, since fire usually burns during many days or months and involves large areas of the landscape. Nevertheless, the majority of wildfires in Europe burn for 48 hours or less, thus fire has fewer opportunities to flip from one fire type to another.

**References:**

Castellnou, M., Pagés, J., Miralles, M., Piqué, M.: Tipificación de los incendios forestales de Catalunña. Elaboración del mapa de incendios de diseño como herramienta para la gestión forestal. Proceedings of the 5th Congreso Forestal Espanñool Ávila, Spain. Available: https://interior.gencat.cat/web/.content/home/030_arees_dactuacio/bombers/foc_forestal/jornades_recerca_cooperacio_internacional/articles_de_recerca_en_foc_forestal/articles_incendis_forestals/2009_Castellnou-et-al_tipificacion-IF-en-CAT_Mapa-incendios-de-diseno_CongrAvila.pdf (last access: 29 July 2024), 2009.

Costa, P., Castellnou, M., Larrañaga, A., Miralles, M., and Kraus, D.: Prevention of Large Wildfires using the Fire Types Concept, Departament de Interior.Generalitat de Catalunya., Cerdanyola del Vall,s, Barcelona, Spain., https://interior.gencat.cat/ca/el_departament/publicacions/proteccio_civil/la_prevencio_dels_grans_incendis_forestals_adaptada_a_l_incendi_tipus/index.html (last access: 29 July 2024), 2011.

Lecina-Diaz, J., Alvarez, A., and Retana, J.: Extreme fire severity patterns in topographic, convective and wind-driven historical wildfires of mediterranean pine forests, PLoS One, https://doi.org/10.1371/journal.pone.0085127, 2014.

Silva, JS., Rego, F., Fernandes, P., Rigolot, E., editors Towards Integrated Fire Management - Outcomes of the European Project Fire Paradox. European Forest Institute Research Report 23. https://efi.int/publications-bank/towards-integrated-fire-management-outcomes-european-project-fire-paradox (last access: 29 July 2024), 2010.

**Line (L) 15. … 'one of the largest wildfires of the last decade'. In what context is this e.g. fires in Spain, fires in the Mediterranean region?**

You are right, we did not specify the location well enough in the abstract. The Jonquera fire was in north-eastern Spain, we have included this information in the abstract.

**Please remove emotive phrases from the manuscript e.g. L25 'massive wildfire', L56 'huge inaccuracies'**

We have revised the text to remove all emotive sentences and rephrase unnecessary nuances.

**L44. You mention health impacts from wildfire particulate matter. It is worth pointing out that gas phase pollutants from wildfire also have health effects as well.**

Thank you, you are right, we have included this point in the sentence.

**L53. Referring to the 'Seiler and Crutzen (1980) method' strikes me as jargon. Technically, it is a fuel consumption method that Seiler and Crutzen (1980) developed.**

Yes, thank you for the observation, we have written another brief paragraph to clarify the method is in comprehensive way.

**Page 2 bottom paragraph. I'm wondering whether the paper below is worth citing to provide a technical definition for what you are referring to as 'fire severity'?**

**E. Keeley. Fire intensity, fire severity and burn severity: A brief review and suggested usage**

**International Journal of Wildland Fire https://doi.org/10.1071/WF07049**

Yes, thank you, we have included a technical definition with your reference (Keeley, 2009) to clarify the meaning of fire severity, together with a suggestion from other reviewer that asks for including a reference for what fire severity is.

**Page 3. L2 and L415. I would remove the phrase 'unprecedented combination of …'. The type of investigation you are conducting is standard practice rather than unprecedented.**

Yes, we have removed "unprecedented combination", but we have highlighted the novelty of the field work data in Spain at least, and the use of litter component in the total fuel load component.

**L115. Moisture content. Is this fine fuel moisture content or something else?**

Yes, that was a mistake. It has been corrected to "Relative humidity".

**Around L130. When you refer to charred trees up to what height level are trees generally charred?**

We have clarified this description in the section "2.3 Field plot data and fire severity estimation", which was split into two different paragraphs with a brief description of the fire severity classification from tree level to plot level following Alvarez et al. (2013).

Reference:

Alvarez, A., Gracia, M., Castellnou, M., and Retana, J.: Variables That Influence Changes in Fire Severity and Their Relationship with Changes Between Surface and Crown Fires in a Wind-Driven Wildfire, For. Sci., 59, 139–150, https://doi.org/10.5849/forsci.10-140, 2013.

**Figure 2. Is this figure adapted or adopted from Alvarez et al. (2012)? If it is adopted you will need copyright permissions to use this figure.**

Thank you for the observation, the figure has been taken and adopted from Alvarez et al. (2012), so, probably we will redraw a new one to convey the same meaning.

**Equation 1. Use multiplication signs rather than the letter x.**

Thank you, this has been changed.

**L233. Log transform for normality. What test did you use for this and what was the result e.g. test statistic and p-value?**

As we have explained in the first response to the reviewer, we have examined the graphical representation of the residuals against the expected values allows us to assess a series of assumptions made about the quality of the model fit: normality, linearity and homoscedasticity. When we transformed the variable into logarithm, the graph of the residuals improved as you can see in the three factor ANOVA of available biomass among fuel types considering the three different layers (crown, shrub and litter) in the two species.

Untransformed available biomass (also in the pdf)

[Figure]

Log-transformed available biomass (also in the pdf)

[Figure]

**L241. What was the required significance level for significant differences?**

The required significance level was 0.05, which corresponds to a 95% confidence level

**Figure 3. Is the log base 10 or base e?**

It is base e.

**Table 2. Is there any reason why nitrous oxide was excluded from your analysis since it is a major greenhouse gas?**

We understand the concern about the lack of nitrous oxide emission values. We only used those gases and pollutants with values from each stratum (crown, shrub, litter) but we did not find emission factors for litter from *Pinus halepensis* and *Quercus suber*. We have added one sentence highlining the importance of having more emission factors available for species especially for nitrous oxide and similar components because of their higher impact on greenhouse phenomenon in the new section **"4.4 Uncertainties in emissions estimates and limitations"**